# A COVID-19 Vaccine for Dogs Prevents Reverse Zoonosis

**DOI:** 10.3390/vaccines10050676

**Published:** 2022-04-24

**Authors:** Eulhae Ga, Yongkwan Won, Jaehyun Hwang, Suyun Moon, Minju Yeom, Kwangsoo Lyoo, Daesub Song, Jeonghee Han, Woonsung Na

**Affiliations:** 1College of Veterinary Medicine, Chonnam National University, Gwangju 61186, Korea; 217057@jnu.ac.kr (E.G.); 218054@jnu.ac.kr (J.H.); seulha0905@jnu.ac.kr (S.M.); 2College of Veterinary Medicine and BK21 FOUR Program, Chonnam National University, Gwangju 61186, Korea; 3Department of Veterinary Pathology, College of Veterinary Medicine and Institute of Veterinary Science, Kangwon National University, Chuncheon 24341, Korea; wyk17@ctcvac.com; 4Research&Development Division R&D Team, CTCVAC Co., Ltd., Hongcheon 25142, Korea; 5College of Veterinary Medicine, Seoul National University, Gwanak-ro, Seoul 08826, Korea; virus1122@korea.ac.kr (M.Y.); sds@snu.ac.kr (D.S.); 6Korea Zoonosis Research Institute, Jeonbuk National University, Iksan 54531, Korea; lks1314@jbnu.ac.kr

**Keywords:** SARS-CoV-2, COVID-19, Coronavirus, vaccine, subunit vaccine, canine, reverse zoonosis, reverse transmission

## Abstract

COVID-19 is caused by severe acute respiratory syndrome virus type 2 (SARS-CoV-2), which can infect both humans and animals. SARS-CoV-2 originated from bats and can affect various species capable of crossing the species barrier due to active mutation. Although reports on reverse zoonosis (human-to-animal transmission) of SARS-CoV-2 remain limited, reverse zoonosis has been reported in many species such as cats, tigers, minks, etc. Therefore, transmission to more animals cannot be ruled out. Moreover, the wide distribution of SARS-CoV-2 in the human population could result in an increased risk of reverse zoonosis. To counteract reverse zoonosis, we developed the first COVID-19 subunit vaccines for dogs, which are representative companion animals, and the vaccine includes the SARS-CoV-2 recombinant protein of whole S1 protein and the receptor-binding domain (RBD). A subunit vaccine is a vaccine developed by purifying only the protein region that induces an immune response instead of the whole pathogen. This type of vaccine is safer than the whole virus vaccine because there is no risk of infection and proliferation through back-mutation of the virus. Vaccines were administered to beagles twice at an interval of 3 weeks subcutaneously and antibody formation rates were assessed in serum. We identified a titer, comparable to that of vaccinated people, shown to be sufficient to protect against SARS-CoV-2. Therefore, the vaccination of companion animals, such as dogs, may prevent reverse zoonosis by protecting animals from SARS-CoV-2; thus, reverse zoonosis of COVID-19 is preventable.

## 1. Introduction

Since the outbreak of coronavirus in 2019, our lives have taken on a new phase. According to the World Health Organization (WHO), as of 17 March 2022, there were 460,280,168 confirmed cases and 6,050,018 deaths. This phase has been predicted because there was a similar situation with several viral zoonotic epidemics. In recent years, important viral infectious diseases have provided clues suggesting that the COVID-19 pandemic includes SARS in 2002–2004, MERS in 2015, which belongs to the same Coronavirus, and pH1N1 influenza in 2009, a representative zoonotic viral infectious disease that does not belong to Coronavirus. Those three diseases are common in the sense that they were caused by RNA viruses that frequently mutate and have a variety of susceptible animal species. All these diseases infect non-human animals first and then spread to humans.

We should note the potential of reverse zoonosis because there were several reports about animal confirmed cases. Coronavirus is a representative virus that has numerous animal species as hosts. It can cause various types of diseases even within the same animal species. The intensity and site of pathogenicity vary according to the cell tropism of the viruses. In the case of dogs, there are some Coronaviruses that cause enteritis (canine coronavirus) or respiratory disease (canine respiratory coronavirus) [1,2]. Coronaviruses occur in various species including feline coronavirus (FCoV), feline infectious peritonitis virus (FIP), bovine coronavirus (BCoV), porcine epidemic diarrhea virus (PED), transmissible gastroenteritis virus (TGE), infectious bronchitis virus (IBV), etc. [2,3,4,5,6,7]. These differences between species and pathogenicity are derived from at least a few parts of sequences [8]. In the case of COVID-19, it is thought that the original host was a bat that passed it onto humans. Moreover, there is about 96% identicality in gene sequences between the viruses in humans and their natural host: the bat [9].

Since SARS-CoV-2 belongs to the RNA virus, it has a high possibility of mutation and the ability to cross species barriers that could result in reverse zoonosis [10]. Reverse zoonosis comprises transmission from humans to animals. Although rare, cases of reverse zoonosis against SARS-CoV-2 have been reported in various animal species such as dogs, cats, tigers, and minks [11,12,13,14].

If reverse zoonosis occurs, it is not only a matter of newly born susceptible species. When viruses infect two or more animal species and move through various environments, there is a possibility that the animal will act as a reservoir and increase the rate of mutations as in the case of swine with influenza A virus, which acts as a mixing vessel [15,16,17,18]. SARS-CoV-2 transmits between species and could evoke a co-infection with other strains of Coronaviruses in a specific host. This could increase the rate of mutation while the virus tries to adapt to various new environments [19,20] and recombinants can occur in the process of replication of two types of viruses in the same cell [21,22,23]. The Coronavirus is well known as a virus that has a very high probability of recombination [24,25,26,27,28]. Even if quarantine against COVID-19 in humans is successful, if infection through various species occurs, it could open a new avenue for virus mutation and reinfection. Therefore, we need to control transmissions not only between human-to-human transmission but also human-to-animal transmission. Companion animals share lifestyles with humans while living on the narrow interspecies surface. For these reasons, companion animals are exposed to reverse zoonosis, and countermeasures are required, such as companion animal vaccines.

Coronaviruses use spike (S) proteins to enter cells binding with the cell’s ACE2 (angiotensin-converting enzyme 2) receptors [29,30,31]. Spike protein is composed of two proteins: S1 and S2. Mainly, S1 protein helps in binding and attachment to host cell receptors, and S2 protein mediates fusion to the cell membrane [32,33,34,35]. Spike proteins induce spike-specific cellular response, and CD^+^8 T cell responses have been observed in the early period [36]. In particular, the receptor-binding domain (RBD) in S1 protein plays a crucial role in successful entry into the host cell, and it is the most dominant antigenic site for inducing SARS-CoV-2-neutralizing antibodies containing the majority of neutralizing epitopes [37]. Therefore, the S protein is a key part in the induction of T-cell responses, as well as protective immunity.

For these reasons, the spike protein is considered a major target protein for vaccine and therapeutic drug development [38]. The World Health Organization (WHO) approved the use of NVX-CoV2373, which is the subunit vaccine that is currently used in humans worldwide. The vaccine uses the SARS-CoV-2 S protein antigen and shows 89.7% effectiveness in participants [39]. The Pfizer-BioNTech (BNT162b2 vaccine) and Moderna vaccines (mRNA-1273 vaccine), approved the use by US Food and Drug Administration (USFDA) and show over 90% effectiveness in people 16 years of age and older, also encode the SARS-CoV-2 S protein [40]. Therefore, we developed a subunit vaccine for companion animals using spike proteins and evaluated its efficacy in the target animals.

Additionally, we figured out that this subunit vaccine does not have a cross-protective ability against other human coronaviruses. Coronaviruses that are prevalent as respiratory diseases in humans include 229E and OC43 viruses [41,42]. Both 229E and OC43 viruses are respiratory viruses in humans, but according to phylogenetic classifications, the 229E virus is an alphacoronavirus and the OC43 virus is a betacoronavirus such as SARS-CoV-2. In particular, the OC43 virus is reported as a virus that has a high S protein sequence similarity with the Coronavirus detected in dogs infected with respiratory disease [43]. We also tested the potential for cross-protection against OC43 and 229E viruses against the COVID-19 vaccine.

## 2. Materials and Methods

### 2.1. Animals

Ten female beagles aged 4–6 months, who received all canine parvovirus, canine distemper virus, canine adenovirus, canine parainfluenza virus, and rabies vaccines (Nobivac Madison, WI. United States) were purchased from Saeronbio [44,45]. They were housed in an isolated cage within the BSL-2 facility at Chonnam National University (Gwangju, Korea) for the study. All animal experiments complied with the current laws of Korea. Animal care and treatment were conducted in accordance with the guidelines established by the Chonnam National University Institutional Animal Care and Use Committee (CNU IACUC-YB-R-2021-98).

### 2.2. Antigen

S1 protein and RBD protein recombinant plasmids were prepared using the pcDNA™3.3-TOPO^®^ vector (ThermoFisher, Waltham, MA. United States. Cat No. K8300-01). The vector was injected into the Chinese Hamster Ovary (CHO) cell to express the protein. After culturing the transformed CHO cells, the supernatant is collected, filtered, and purified by column loading (GE healthcare, AKTA prime plus). After purification, the protein is identified by SDS-PAGE and Western Blotting. The purified antigen is used as the vaccine antigen after checking the concentration using Nanodrop (ThermoFisher, Waltham, MA. United States. AZY2017596) (Figure 1).

### 2.3. Vaccination

A vaccine antigen consists of SARS-CoV-2 spike 1 (S1) protein and the receptor-binding domain (RBD) protein manufactured by CTCVAC Co., Ltd, Hongcheon, Korea. (CTCVAC FCoV-19, lot# CCVa-2101), and the method is described in Figure 1. The vaccine contains an adjuvant of 10% Montanide gel (PR02, Seppic) or stimulant of monophosphoryl lipid A (MPL, TLR4 agonist; Sigma-Aldrich, Darmstadt, Germany.). MPL mixed with DW containing 0.2% trimethylamine was heated at 70 °C for 30 s, then sonicated for 30 s, and these steps were repeated twice (Table 1).

Test vaccine formulation and vaccination schedules are shown below in Table 1 and Table 2 and Figure 2.

### 2.4. Cells and Viruses

Human colon adenocarcinoma cells (HCT-8, KCLB, KCLB No. 10,244), human lung fibroblast cells (MRC-5, KCLB, KCLB No. 10,171), and African green monkey kidney epithelial cells (VERO C1008, ATCC, CRL-1586) were obtained from the Korean Cell Line Bank (KCLB) and American Type Culture Collection (ATCC), respectively. The cells were grown at 37 °C in air enriched with 5% CO_2_(carbon dioxide) and Dulbecco’s modified Eagle’s medium (DMEM, CORNING 10-013-CV) or Roswell Park memorial institute 1640 (RPMI, WELGENE LM011-51) medium supplemented with 10% fetal bovine serum (FBS) and 1% penicillin (100 units/mL)-streptomycin (100 μg/mL)-2mM L-glutamine. OC43, 229E, and SARS-CoV-2 viruses were grown in HCT-8, MRC-5, and Vero cells respectively.

### 2.5. Groups

Beagles were divided into a negative control group and two experimental groups. Experimental group 1 was administered with a vaccine-containing adjuvant. Experimental group 2 was administered with a vaccine containing both the adjuvant and stimulant. In the negative control group, PBS was used instead of the vaccines. Each vaccine was injected through the subcutaneous (S.C) route on the first day, and the second injection was performed 3 weeks later in the same manner (Table 2). Blood collection was performed on the day of first and secondary vaccination and 2 weeks after secondary vaccination. Bodyweight and body temperature were measured by four veterinarians observing general clinical signs and local side effects during the experiment (Figure 2).

### 2.6. SN Test

For the serum neutralization test, fresh serum was collected from dogs. The serum was stored at −70 °C. Cells were prepared in 96-well plates at a dilution of 1×10^5^/mL with a cell culture medium. All test sera were heated at 56 °C for 30 min and were diluted 2-fold serially in flat-bottom 96-well microtiter plates using PBS as diluents. In each diluted sera, viruses were added with a 1:1 ratio after dilution to 100TCID_50_/well. After mixing, plates were incubated at 37 °C for 1 h. Moreover, the mixed sera were dispensed on the prepared corresponding cells. Then, incubation was performed with 5% CO_2_ and 33 °C (OC43, 229E) or 37 °C (SARS-CoV-2) temperature in an incubator for 3–5 days. When the virus titer reaches 100TCID_50_/well, the neutralizing ability is measured by observing the cytopathogenic effect (CPE) [46,47,48,49]. The experiment was performed in duplicates.

## 3. Results

### 3.1. Side Effects of Vaccination

There was no specific side effect after vaccination. No systemic side effects such as fever, neurological symptoms, anorexia, and lethargy were observed. No local side effects such as swelling, redness, itching, granuloma, and suppuration were observed. The normal body temperature range for dogs is 38–39 °C, and in the cases of young dogs, the range of body temperature could be slightly wider or higher [50]. No abnormal change in body temperature was observed during this experiment (Figure 3a).

Beagle grows rapidly up to 4–6 months, and the growth rate gradually decreases, reaching the size of an adult dog at 12 months old. During this experiment, the growth rate was about 10–15% per month following the general growth curve and no abnormal weight loss was observed [50]. No abnormal change in body weight was observed during this experiment (Figure 3b).

### 3.2. SN Test

From the results of the serum neutralization test, significant antibody values began to appear 2 weeks after the second vaccination for COVID-19. Much higher amounts of neutralizing antibodies were found in Group 2 in which stimulants were added. We identified a titer shown to be sufficient in protecting against SARS-CoV-2, comparable to that of vaccinated people. According to the previous study, average serum neutralization titers were 80 for paucisymptomatic patients, 160 for symptomatic patients, 40 for primary vaccinees, and 160 for secondary vaccinees [51]. Group 2 was found to have higher titers than the vaccinated people in both the first and second vaccinations. Both Groups 1 and 2 were found to have higher titers than the symptomatic patients when the second vaccination was completed (Table 3, Figure 4).

In the SN test, we judged the presence of virus in the wells with the presence of CPE in the cells, and the micrographs are as follows (Figure 5). In the experiment to observe the cross-protective ability against OC43 and 229E viruses, CPE was observed in all experimental groups in which the antibody and viruses were reacted. Therefore, it was judged that there was no cross-protective ability.

## 4. Discussion

In both vaccinated groups, antibody-forming rates significantly increased after the second vaccination over the level of neutralizing antibody expressed in the vaccinated or infected person. The group containing the stimulant showed 6-times higher antibody-forming rates than the group without the stimulant, because the stimulant induces the immune responses of the monocytes and enhances innate immunity [52]. The induction of antibody-forming rate in this vaccine that used S protein and RBD is consistent with the results of previous studies showing that the S protein and especially RBD play a major role in the induction of cellular immunity [53,54].

The RBD regions of SARS-CoV-2 and SARS-CoV share about 70% sequences [55], so there is a report that cross-protection against SARS-CoV was successful with mRNA vaccine using RBD of SARS-CoV-2 [54]. Therefore, we tested the possibility of cross-protection against OC43 and 229E viruses, which infect humans generally and routinely than compared to the pandemic virus-like SARS-CoV. As a result, the antibody induced through the RBD and S1 proteins of SARS-CoV-2 had no cross-protective ability against OC43 and 229E viruses, and it was consistent with the results of previous studies conducted in humans [56].

Although no virus infection test was performed, the protective ability against infection can be sufficiently predicted by measuring the neutralizing antibody value through the SN test. If this animal subunit vaccine is vaccinated in companion animals, it is expected to help reduce the possibility of reverse zoonosis. It is necessary to confirm these conjectures by conducting additional experiments in various companion animals and livestock breeds such as cats, minks, and pigs in the future.

## Figures and Tables

**Figure 1 vaccines-10-00676-f001:**
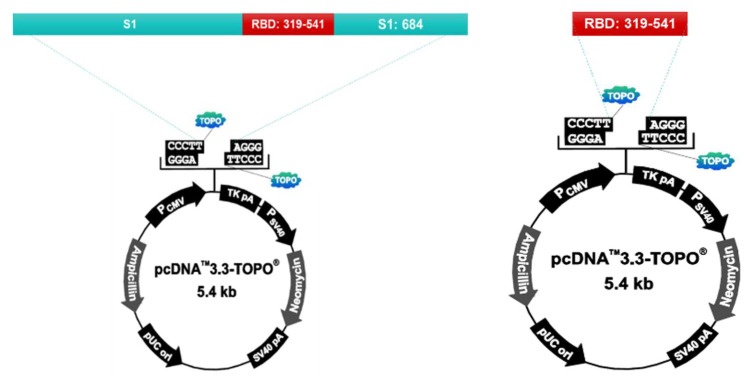
S1 protein and RBD protein recombinant plasmids were prepared using the pcDNA™3.3-TOPO^®^ vector (ThermoFisher, Waltham, MA. United States. Cat No. K8300-01). Chinese Hamster Ovary(CHO) cells were used for protein expression. After culturing the transformed CHO cells, the supernatant is collected, filtered, and puri-fied by column loading (GE healthcare, AKTA prime plus).

**Figure 2 vaccines-10-00676-f002:**
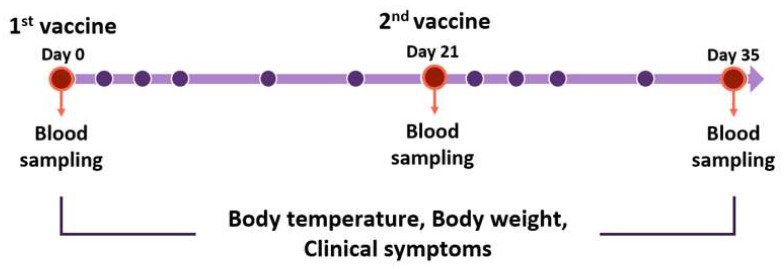
Vaccination schedule for Beagles. Beagles were moored for a week before the experiment. Vaccines were inoculated on day 0, 21, and 35. Blood samplings were performed before vaccination. Body temperature, body weight, and clinical symptoms were measured. The red circles mean the blood sampling date. The purple circles mean the date of checking body temperature, weights, and symptoms for clinical monitoring. All experiments and monitoring were performed by four veterinarians.

**Figure 3 vaccines-10-00676-f003:**
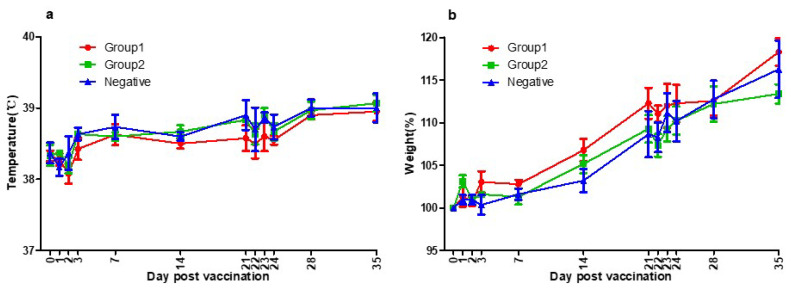
(**a**) Three groups’ temperatures of the day post-vaccination. Group1 (●), Group2 (■), and Negative Group (▲) showed the same patterns of body temperature. They were all within the normal range. As a result of temperature measurements, there was no side effect of the vaccines. (**b**) Three groups’ body weights of the day post-vaccination. Group1 (●), Group2 (■), and Negative Group (▲) showed the same patterns of growth rate. They were all within the normal range. As a result of bodyweight measurements, there was no side effect of the vaccines.

**Figure 4 vaccines-10-00676-f004:**
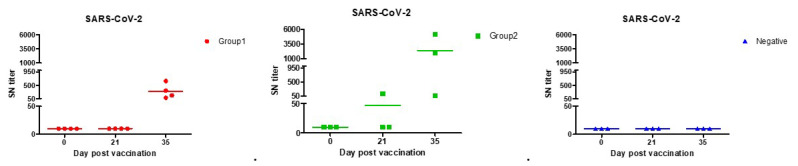
SARS-CoV-2 neutralization antibody forming rates with each of the beagles. Both Group 1 (●) and Group 2 (■) had sufficient antibodies against SARS-CoV-2 infection in second vaccination. According to previous studies, the antibody produced in vaccinated persons, asymptomatic, and symptomatic patients had an SN titer of up to 160.

**Figure 5 vaccines-10-00676-f005:**
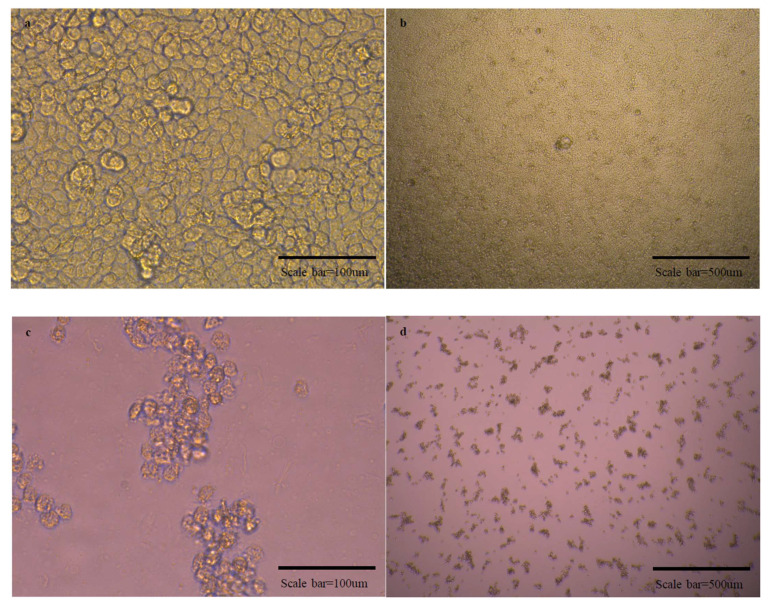
Micrographs of cells in normal state and cells with cytopathic effects due to viral infection. There are micrographs of SARS-CoV-2/Vero (**a**–**d**), OC43/HCT-8 (**e**–**h**), and 229E/MRC-5 (**i**–**l**) in order. There are high magnification micrographs (200×) of normal state (**a**,**e**,**i**) and cytopathic effect (**c**,**g**,**k**) cells. There are low magnification micrographs (40×) of normal state (**b**,**f**,**j**) and cytopathic effect (**d**,**h**,**l**) cells. In the SN test conducted additionally using OC43 and 229E viruses, cross-protection was not shown.

**Table 1 vaccines-10-00676-t001:** Materials used in the construction of vaccines are in the table. S1 protein and RBD protein were used as antigens, Montanide gel as an adjuvant, and MPL as a stimulant in the ratio as above.

Per Dose		
Antigen	S1 (Spike Protein 1)	60 μg
	RBD (Receptor binding domain)	60 μg
Adjuvant	Montanide gel (PR02)	10%(vol)
Stimulant	MPL (Sigma)	50 μg

**Table 2 vaccines-10-00676-t002:** Experimental design for immunization in beagle dogs offered by COVID-19 subunit vaccines. Vaccines used for Group 1 were composed of SARS-CoV-2′s S1, RBD protein, and adjuvants. Vaccines used for Group 2 were composed of SARS-CoV-2′s S1, RBD protein, and adjuvants and stimulants were added. Vaccines or PBS were inoculated by subcutaneous route.

Group	Immunization Route	Vaccine	Number of Dogs
Group 1	S.C	FCoV-19	4
Group 2	S.C	FCoV-19+stl ^1^	3
Unvaccinated	S.C	PBS	3

^1^ FCoV-19 vaccines containing stimulants.

**Table 3 vaccines-10-00676-t003:** Comparison table of serum virus neutralization test titer between the tested group, commercially available vaccine group, and paucisymptomatic or symptomatic patient group. Group 2 was found to have higher titers than the vaccinated people in both the first and second vaccination. Both Groups 1 and 2 were found to have higher titers than the symptomatic patients when the second vaccination was completed.

	Paucisymptomatic Patients	Symptomatic Patients		Vaccinated People	Group 1	Group 2	Negative
SN titer	1:80	1:160	SN titer (1st)	40	10	47	10
			SN titer (2nd)	160	226	833	10

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
