# Peer review of "A COVID-19 Vaccine for Dogs Prevents Reverse Zoonosis"

_vaccines, 2022, doi:10.3390/vaccines10050676_

Round 1

Reviewer 1 Report

By developing vaccination strategies against COVID-19 for human companians, the current manuscripts pursues an important strategy to limit not only the spreading, but also the evolutionary drift of this human pathogen. Thus, successful vaccination of dogs might indeed consist an important measure to tackle this world-wide pandemic. Although the pursued strategy is straight-forward, there are, however, a number of significant drawbacks to enable a sincere judgment regarding the potential benefit of the described research:

  • The neutralization capacity of serum-containing antibodies was assessed at three time-points, i.e. before the vaccination started, at the time-point of the 2nd vaccination (i.e. 21 days post 1st vaccination, and at day 35 (14 days after the second vaccination). This is most likely the time when peak-amounts of COVID-19 antibodies are expected, suggesting the best possible neutralization effect. This is indeed sufficient to demonstrate a potential immune-response and to collect antibodies for various purposes, however, has only limited significance to demonstrate protective immunity against a potential agent. It is therefore important to follow-up neutralization capacity for an extended period of time after vaccination.
  • Materials and Methods are very superficial, sometimes lacking significant information and not really informative. Most importantly: (i) Production and purification of the antigen: Although from vectors shown it is indicative that the antigens are produced in mammalian cells, the source of production should be mentioned, since there are large differences regarding integrity and modifications of proteins produced with different production systems. In addition, there is no information about the purification/purity of the recombinant proteins. (ii) There is no indication about the incubation time of SN-Tests, i.e. whether the antisera tested are indeed protective or just capable to inhibit incubation for a limited time-frame. (iii) Table 3: Were these experiments performed in parallel?

Minor points:

Lane 56ff: These differences between species … This sentence does not make sense. Are there only a few amino acid differences or are only a few aa responsible for differences in pathogenicity or what? In addition, differences regarding the pathogenicity should be described in this paragraph.

Lane 96ff: Although potential differences and phylogenetic classifications are important, it would be of interest to draw a conclusion towards the described strategy.

Author Response

Respond to Reviewer

  1. This is indeed sufficient to demonstrate a potential immune-response and to collect antibodies for various purposes, however, has only limited significance to demonstrate protective immunity against a potential agent. It is therefore important to follow-up neutralization capacity for an extended period of time after vaccination.
  • As reviewer's advice, many questions have been raised about the duration of vaccines recently. In this experiment, we have figured out only about the protective ability of vaccine. However, we agree with the need of further study of the vaccine’s duration with the extended long period.
  1. Production and purification of the antigen: Although from vectors shown it is indicative that the antigens are produced in mammalian cells, the source of production should be mentioned, since there are large differences regarding integrity and modifications of proteins produced with different production systems. In addition, there is no information about the purification/purity of the recombinant proteins.
  • We agree with reviewer’s advice. It seems that the information of antigen production was insufficient. We described the information of the transformed cells, protein purification and enrichment on the methods.

Antigen

S1 protein and RBD protein recombinant plasmids were prepared using pcDNA™3.3-TOPO® vector (ThermoFisher, Cat No. K8300-01). The vector was injected into Chinese Hamster Ovary(CHO) cell to express the protein. After culturing the transformed CHO cells, the supernatant is collected, filtered, and purified by column loading(GE healthcare, AKTA prime plus). After purification, the protein is identified through SDS-PAGE and Western Blotting. The purified antigen is used as the vaccine antigen after checking the concentration using Nanodrop(Thermo, AZY2017596).

  1. There is no indication about the incubation time of SN-Tests, i.e. whether the antisera tested are indeed protective or just capable to inhibit incubation for a limited time-frame.
  • As reviewer’s advice, we added a phrase regarding incubation time during the SN test. After inoculation with the virus&serum reaction solution, the cells were incubated for 3-5 days until the CPE effect of the virus appeared. The serum neutralization test provides on information of protective level of antibody that indeed neutralizing the pathogen. Therefore it can be interpreted as an effective defense against the virus, rather than a short term blocking of the virus’s proliferation.

SN test. For the serum neutralization test, fresh serum was collected from dogs. The serum was stored at -70℃. Cells were prepared in 96-well plates at dilution of 1*10^5/ml with cell culture medium. All test sera were heated at 56℃ for 30 minutes and were diluted 2-fold serially in flat-bottom 96-well microtiter plates using PBS as diluents. In each diluted sera, viruses were added with 1:1 ratio after dilution to 100TCID50/well. After mixing, plates were incubated at 37℃ for 1 hour. And the mixed sera were dispensed on the prepared corresponding cells. Then incubating was performed with 5% CO2 and 33℃(OC43, 229E) or 37℃(SARS-CoV-2) temperature in an incubator 3-5 days. When the virus titer reaches 100TCID50/well, the neutralizing ability is measured by observing the cytopathogenic effect (CPE)[45-48].

  1. Table 3: Were these experiments performed in parallel?
  • We performed the SN test by duplicates and recorded the geometric average. As reviewer’s advice, we added a comment about parallel performing the experiments.

When the virus titer reaches 100TCID50/well, the neutralizing ability is measured by observing the cytopathogenic effect (CPE)[45-48]. The experiment was performed in duplicates.

  1. About minor points
  • We agree with the reviewer's advice. We corrected the sentence and relocated the phylogenetic explanation.

These differences between species and pathogenicity are derived from at least a few part of sequences[8]. In the case of COVID-19, it is thought that the original host was a bat and passed it on to humans. And there is about 96% identical in gene sequence between the viruses in human and its natural host, bat[9].

Additionally, we figure out this subunit vaccine doesn’t have the cross-protective ability against other human coronaviruses. Coronaviruses that are prevalent as respiratory diseases in humans include the 229E and OC43 viruses[41, 42]. Both 229E and OC43 viruses are respiratory viruses in humans but according to the phylogenetic classification, 229E virus is an alphacoronavirus and OC43 virus is a betacoronavirus such as SARS-CoV-2. In particular, the OC43 virus is reported as a virus which has high S protein sequence similarity with the Coronavirus detected in dog infected with respiratory disease[43]. We also tested the potential for cross-protection against OC43 and 229E viruses against the COVID-19 vaccine.

Reviewer 2 Report

Ga et al. demonstrated the efficacy of a COVID-19 vaccine for dogs in preventing the potential zoonosis. The authors focused on the representative companion animal, the dog, which is reasonable considering the primary mode of transmission of this infectious disease.

Some of the critical data sets to support the conclusions are missing. In addition, presenting additional data on binding antibodies is recommended.

The manuscript can be improved if the authors address the following points.

  1. This manuscript uses the same strategy as NVX-CoV2373, whose efficacy is reported (Dunkle LM et al., NEJM, 2022 and elsewhere). Rather than mRNA vaccines which use distinct technology from this manuscript, this subunit vaccine should be discussed in the introduction section.

  1. On page 2, line 84, “S1 protein plays crucial roles in successful entry to host cell, and it is the most dominant antigenic site for inducing SARS-CoV-2 neutralizing antibodies containing the majority of neutralizing epitope[36]. Therefore, S protein is the key part in the induction of T-cell responses, as well as protective immunity”, is found. However, induction of neutralizing antibodies does not guarantee T cell responses. Therefore, please cite relevant references as appropriate and modify the statement.

  1. In Figure 2, it is unclear what the purple closed circles stand for. Please clarify.

  1. In Table 3, “Vaccinated people” is found, but were these data obtained from dogs?

  1. In Figure 4, In the legend, “SARS-CoV-2/Vero cell CPE; OC43/HCT-8 cell CPE; 229E/MRC-5 cell CPE.” is found, but no data is visible in the pdf available for the reviewer.

  1. Figure 5. “OC43/HCT-8(e, f, g, h) and 229E/MRC-5(i, j, k, l) in order. There are high magnification micrographs(200x) of normal state(a, e, i) and cytopathic effect(c, g, k ) cells. There are low magnification micrographs(40x) of normal state(b, f, j) and cytopathic effect(d, h, l ) cells. In the SN test conducted additionally using the OC43 and 229E viruses, cross-protection was not shown.” is found, but no micrographs for e, f, g, h, i, j, k, l are found in the pdf. Since it is not available for the reviewer, it is impossible to judge if this vaccine induced cross-protection or not. Please update the manuscript.

  1. The advantage of the use of the mixture of S1 and RBD proteins can be better explained if data on binding antibody titers against S1 or RBD protein determined by ELISA are presented.

Author Response

Respond to Reviewer

  1. Rather than mRNA vaccines which use distinct technology from this manuscript, this subunit vaccine should be discussed in the introduction section.
  • We agree with the reviewer’s advice and corrected the sentence. We replaced the sentence by comparing it with NVX-CoV2373, the same subunit type vaccine, rather than mRNA vaccine, which is a vaccine of different technology.

For these reasons, the spike protein is considered as a major target protein for vaccine and therapeutic drug development[37]. World Health Organization (WHO) approved the use of the NVX-CoV2373 which is the subunit vaccine that are currently used in human worldwide. The vaccine uses the SARS-CoV-2 S protein antigen and shows 89.7% effective in participants[38]. The Pfizer-BioNTech (BNT162b2 vaccine) and Moderna vaccine (mRNA-1273 vaccine), approved the use by US Food and Drug Administration (USFDA) and show over 90% effective in people 16 years of age and older, also encode the SARS-CoV-2 S protein[39]. Therefore, we developed a subunit vaccine for companion animals using spike protein and evaluated its efficacy in target animals.

  1. Induction of neutralizing antibodies does not guarantee T cell responses. Therefore, please cite relevant references as appropriate and modify the statement.

  • We agree with the reviewer’s opinion that neutralizing antibodies does not guarantee T cell responses. We cited appropriate reference and correct the statement(Moss, P. The T cell immune response against SARS-CoV-2. Nat Immunol 23, 186–193 (2022). https://doi.org/10.1038/s41590-021-01122-w).

Spike protein is composed of two proteins, S1 and S2. Mainly, S1 protein helps in binding and attachment to host cell receptor and S2 protein mediates fusion to the cell membrane[32-35].Spike protein induces spike-specific cellular response and CD+8 T cell response have seen in early period[36]. Especially receptor-binding-domain(RBD) in S1 protein plays crucial roles in successful entry to host cell and it is the most dominant antigenic site for inducing SARS-CoV-2 neutralizing antibodies containing the majority of neutralizing epitope[37]. Therefore, S protein is the key part in the induction of T-cell responses, as well as protective immunity.

  •  

  1. In Figure 2, it is unclear what the purple closed circles stand for. Please clarify.
  • As reviewer’s advice, we corrected the figure and clarified what the purple closed circle stand for in figure explanation.
Please see the attachment.

  1. In Table 3, “Vaccinated people” is found, but were these data obtained from dogs?
  • We agree with the reviewer’s opinion. Our experiments were conducted in dogs. To determine whether the antibody titer derived from dogs is sufficient for protection, we refer to human cases that antibody levels were reported from the infected or vaccinated people. Since the SARS-CoV-2 mainly infects humans, the criteria of sufficient SN titer for protection was set as the case in humans.

  1. In Figure 4, In the legend, “SARS-CoV-2/Vero cell CPE; OC43/HCT-8 cell CPE; 229E/MRC-5 cell CPE.” is found, but no data is visible in the pdf available for the reviewer. In Figure 5, no micrographs for e, f, g, h, i, j, k, l are found in the pdf. Since it is not available for the reviewer, it is impossible to judge if this vaccine induced cross-protection or not. Please update the manuscript.

  • We re-uploaded related legends and figures. There seems to have been a problem in the process of converting and completing the file. Additionally, related legends and figures are attached.

  1. The advantage of the use of the mixture of S1 and RBD proteins can be better explained if data on binding antibody titers against S1 or RBD protein determined by ELISA are presented.

  • We agree reviewer’s advice. But It is not possible to prepare an experiment to measure the production of the S1 protein and the RBD-specific antibody respectively within this review period. However, we confirmed sufficient antibody forming rate to protect from infection through the SN test. And the immune-inducing ability for each protein is already known. In previous studies, RBD was found to be more effective in inducing antisera than S1 protein(He, Y., Zhou, Y., Liu, S., Kou, Z., Li, W., Farzan, M., & Jiang, S. (2004). Receptor-binding domain of SARS-CoV spike protein induces highly potent neutralizing antibodies: implication for developing subunit vaccine. Biochemical and biophysical research communications, 324(2), 773-781.). Based on experimental animal ethics, we didn’t set the group verified through previous studies. Therefore, we developed a vaccine by combining RBD and S1 protein without verification process between two proteins.

Round 2

Reviewer 1 Report

Fine with me.

Author Response

Respond to Reviewer

  1. Research design can be improved.
  • We agree with the reviewer’s advice. There was insufficient explanation of the experimental methods and results. Especially, the explanation about the results of the cross-protection test was poorly explained, so we corrected it as follows(Paragraph 2, 3).

  1. Description of methods can be improved.

  • We agree with the reviewer’s advice and corrected the manuscript. We cited figure 1, 2 and table 1, 2 in the manuscript after the corresponding explanation.

S1 protein and RBD protein recombinant plasmids were prepared using pcDNA™3.3-TOPO® vector (ThermoFisher, Cat No. K8300-01). The vector was injected into Chinese Hamster Ovary(CHO) cell to express the protein. After culturing the transformed CHO cells, the supernatant is collected, filtered, and purified by column loading(GE healthcare, AKTA prime plus). After purification, the protein is identified through SDS-PAGE and Western Blotting. The purified antigen is used as the vaccine antigen after checking the concentration using Nanodrop(ThermoFisher, AZY2017596) (Figure 1).

Vaccine antigen consists SARS-CoV-2 spike 1(S1) protein and the receptor binding domain(RBD) protein manufactured by CTCVAC Co., Ltd (CTCVAC FCoV-19, lot# CCVa-2101) and the method is described in figure 1. The vaccine contains adjuvant of 10% Montanide gel(PR02, Seppic) or stimulant of monophosphoryl lipid A(MPL, TLR4 agonist; Sigma-Aldrich). MPL mixed with DW containing 0.2% trimethylamine was heated at 70 °C for 30 seconds, then sonicated for 30 seconds, and these steps were repeated twice (Table 1).

Beagles were divided into a negative control group and two experimental groups. Experimental group 1 was administered with vaccine containing adjuvant. Experimental group 2 was administered with vaccine containing both the adjuvant and stimulant. In the negative control group, PBS was used instead of the vaccines. Each vaccine was injected through the subcutaneous (S.C) route on the first day, and the second injection was performed 3 weeks later in the same manner (Table 2).

Blood collection was performed on the day of first and secondary vaccination and 2 weeks after secondary vaccination. Body weight and body temperature are measured by four veterinarians observing general clinical signs and local side effects during experiment(Figure 2).

  1. Presentation of results can be improved.

  • We agree with the reviewer’s advice and corrected the manuscript. We cited figure 4 and table 3 in the manuscript after the corresponding explanation. We explained for figure 5 and cited the figure 5 after the explanation. And we mentioned about negative data of additional experiments to check the cross-protection.

As results of the serum neutralization test, significant antibody values began to appear 2 weeks after the second vaccination for COVID-19. Much higher amounts of neutralizing antibody were found in group 2, which stimulants were added. We identified a titer shown enough to protect against SARS-CoV-2, comparable to that of vaccinated people. According to the previous study, average serum neutralization titers were 80 for paucisymptomatic patients, 160 for symptomatic patient, 40 for primary vaccinees, and 160 for secondary vaccinees[1]. Group 2 was found to have higher titers than the vaccinated people in both the first and second vaccination. Both group 1 and 2 were found to have higher titers than the symptomatic patients when the second vaccination was completed(Table 3, Figure 4).

In the SN test, we judged the presence of virus in the wells through the presence of CPE in the cells, and the micrographs are as follows(Figure 5). In the experiment to observe the cross-protective ability against OC43 and 229E viruses, CPE was observed in all experimental groups in which the antibody and the viruses were reacted. Therefore it was judged that there was no cross-protective ability.

All the authors have seen the manuscript and have agreed to the submission decision. Thank you very much for your kind consideration in advance 

Sincerely yours,

Woonsung Na

College of Veterinary Medicine, Chonnam National University, Gwangju, 61186

Republic of Korea

Tel: +82 62-530 2814

Reviewer 2 Report

The missing data (Figure 5) are now available for the reviewer. However, all the figures and tables, including this Figure 5 should be cited and explained in the text as appropriate. Please check the manuscript carefully to ensure Table 3, Figures 4-5 are quoted as appropriate. Figure 5 is not mentioned in the result section. In addition, "In the SN test conducted additionally using the OC43 and 229E viruses, 231 cross-protection was not shown." is found in the figure legend. Still, the negative data should be presented as supplementary data and explained in the result section, as this is one of the main findings of this manuscript.

Author Response

Respond to Reviewer

  1. The figures should be cited and explained in the test as appropriate.
  • We agree with the reviewer’s advice and corrected the manuscript. We cited figure 1, 2, 4 and table 1, 2, 3 in the manuscript after the corresponding explanation. And as the reviewer's advice, we explained for figure 5 and cited the figure 5 after the explanation.

S1 protein and RBD protein recombinant plasmids were prepared using pcDNA™3.3-TOPO® vector (ThermoFisher, Cat No. K8300-01). The vector was injected into Chinese Hamster Ovary(CHO) cell to express the protein. After culturing the transformed CHO cells, the supernatant is collected, filtered, and purified by column loading(GE healthcare, AKTA prime plus). After purification, the protein is identified through SDS-PAGE and Western Blotting. The purified antigen is used as the vaccine antigen after checking the concentration using Nanodrop(ThermoFisher, AZY2017596) (Figure 1).

Vaccine antigen consists SARS-CoV-2 spike 1(S1) protein and the receptor binding domain(RBD) protein manufactured by CTCVAC Co., Ltd (CTCVAC FCoV-19, lot# CCVa-2101) and the method is described in figure 1. The vaccine contains adjuvant of 10% Montanide gel(PR02, Seppic) or stimulant of monophosphoryl lipid A(MPL, TLR4 agonist; Sigma-Aldrich). MPL mixed with DW containing 0.2% trimethylamine was heated at 70 °C for 30 seconds, then sonicated for 30 seconds, and these steps were repeated twice(Table 1).

Test vaccine formulation and vaccination schedules are figured out below table1, 2 and figure 2.

Groups. Beagles were divided into a negative control group and two experimental groups. Experimental group 1 was administered with vaccine containing adjuvant. Experimental group 2 was administered with vaccine containing both the adjuvant and stimulant. In the negative control group, PBS was used instead of the vaccines. Each vaccine was injected through the subcutaneous (S.C) route on the first day, and the second injection was performed 3 weeks later in the same manner(Table 2). Blood collection was performed on the day of first and secondary vaccination and 2 weeks after secondary vaccination. Body weight and body temperature are measured by four veterinarians observing general clinical signs and local side effects during experiment(Figure 2).

SN test. As results of the serum neutralization test, significant antibody values began to appear 2 weeks after the second vaccination for COVID-19. Much higher amounts of neutralizing antibody were found in group 2, which stimulants were added. We identified a titer shown enough to protect against SARS-CoV-2, comparable to that of vaccinated people. According to the previous study, average serum neutralization titers were 80 for paucisymptomatic patients, 160 for symptomatic patient, 40 for primary vaccinees, and 160 for secondary vaccinees[1]. Group 2 was found to have higher titers than the vaccinated people in both the first and second vaccination. Both group 1 and 2 were found to have higher titers than the symptomatic patients when the second vaccination was completed (Table 3, Figure 4).

In the SN test, we judged the presence of virus in the wells through the presence of CPE in the cells, and the micrographs are as follows(Figure 5).

  1. About the SN test using OC43 and 229E viruses, the negative data should be presented as supplementary data and explained in the result section, as this is one of the main findings of this manuscript

  • As the reviewer's advice, we mentioned about negative data of additional experiments to check the cross-protection.

In the experiment to observe the cross-protective ability against OC43 and 229E viruses, CPE was observed in all experimental groups in which the antibody and the viruses were reacted. Therefore it was judged that there was no cross-protective ability.

All the authors have seen the manuscript and have agreed to the submission decision. Thank you very much for your kind consideration in advance 

Sincerely yours,

Woonsung Na

College of Veterinary Medicine, Chonnam National University, Gwangju, 61186

Republic of Korea

Tel: +82 62-530 2814
